# Enhanced Electrical Properties of Polyethylene-Graft-Polystyrene/LDPE Composites

**DOI:** 10.3390/polym12010124

**Published:** 2020-01-05

**Authors:** Shuwei Song, Hong Zhao, Zhanhai Yao, Zhiyu Yan, Jiaming Yang, Xuan Wang, Xindong Zhao

**Affiliations:** 1Wang Da-Heng Collaborative Innovation Center, Heilongjiang Provincial Key Laboratory of Quantum Manipulation & Control, Harbin University of Science and Technology, Harbin 150080, China; ssw@hrbust.edu.cn; 2Key Laboratory of Engineering Dielectrics and Its Application, Harbin University of Science and Technology, Harbin 150080, China; jmyang@hrbust.edu.cn (J.Y.); topix@sina.com (X.W.); hlgzxd@163.com (X.Z.); 3State Key Laboratory of Polymer Physics and Chemistry, Changchun Institute of Applied Chemistry Chinese Academy of Sciences, Changchun 130022, China; yaozh@ciac.jl.cn; 4Zhongtian Technology Submarine Cables Co., Ltd., Nantong 226010, China; yzy510429304@126.com

**Keywords:** cables, conductivity, electric fields, high voltage direct current (HVDC) insulators, nanocomposite

## Abstract

Nanocomposite dielectrics show a great potential application in high voltage direct current cables for their obvious improvements in electrical properties. In the present manuscript, nanocomposite composed of low-density polyethylene and nanoscale polystyrene particles is studied by using low-density polyethylene grafted with polystyrene molecule. Fourier-transform infrared spectra reveal successful grafting of the polystyrene molecule onto the low-density polyethylene chain and the scanning electron microscope image shows the homogeneously dispersed nanoscale polystyrene particles. The presence of the polystyrene nanoparticles obviously improves the dielectric properties, such as the direct current breakdown strength and space charge inhibition. The conductivity and thermally stimulated current characteristics imply the deep traps in the composite increase obviously. Density functional theory calculation reveals that the grafted polystyrene can accommodate both shallow and deep electron carriers, and the depth of the hole traps are as deep as 2.07 eV.

## 1. Introduction

The modern power system necessitates the efficiency of long-distance power transmission since the worldwide demand for electrical energy increases rapidly. In addition, power transmission should reduce environmental impact as much as possible for populated or environmentally sensitive areas [1]. All these are calling for high voltage direct current (HVDC) transmission lines, among which extruded direct current cables have played an active role [2]. These days, polymeric HVDC cables are becoming increasingly important, and the range of their applications has been prominently expanded [3,4,5,6,7]. In spite of the prominent success with polymeric HVDC cables, insulating materials used for HVDC cables are still suffering from annoying limitation and the critical one of them is the space charge accumulation problem.

The space charge accumulation would have a dramatic effect on the electric field distribution, leading to local electric field distortion [8,9,10]. As a result, the space charge accumulation can degrade the cable dielectrics from a long-term point of view. In the case of polarity reversal for line commutated converter (LCC) lines, the destructiveness of accumulated space charge can become even more obvious. Homo charges before the polarity reversal become hetero charges afterward, resulting in dramatic distortion of the electric field around these accumulated charges. Since the voltage source converter (VSC) has a high degree of flexibility and the development of this technology is likely to lead to more efficient and cheaper use of VSC-HVDC [11], the polarity reversal problem is not important. In any case, an endeavor has been made to suppress the space charge accumulation. One effective method is to incorporate nanoscale fillers in a polymeric matrix.

Since T.J. Lewis proposed the concept of nanodielectrics [12], nanocomposites have been explored as insulation materials [13,14,15,16,17,18,19,20]. Comparing to the conventional fillers, Nelson et al. proved experimentally the effectiveness of nanofillers in improving the electrical properties [21]. From then on, various kinds of inorganic nanoparticles have been exploited to suppress space charge accumulation in polymeric insulation materials, like silica [22,23,24,25], magnesia [26,27], titanium oxide [28,29], alumina [30,31], zinc oxide [32,33], montmorillonite [34], zeolite [35], boron nitride [36], etc. The problem with nanofillers is that they would aggregate to form microscopic clusters.

As pointed out in Ref. [37], nanodielectrics can possibly unleash their full potential only if the fillers are distributed evenly throughout the host material. To ensure the homogeneous distribution of nanofillers, surface functionalization seems necessary during the synthesis process. For surface treatment by physical methods, the surfactant is assessed by the tendency to be adsorbed around the interfaces, and the surfactant concentration at the boundary is determined by both its structure and the nature of the two phases defining the interface. For chemical methods, the coupling agent should be carefully chosen in order to achieve favorable homogeneous dispersion of nanofillers in the polymer matrix. Therefore, the surfactant or coupling agent should not always be effective [38]. The technical difficulties still lie in the way to the effective modification of the nanoparticle surface, and the cost of mass-producing high-performance nanocomposites is too high. In addition, even a tiny amount of agglomerated clusters can cause a blocking of the metal wire when passing through the finely-meshed filtrating screen in the cable manufacturing process.

Our former studies show that nanoscale polystyrene (PS) particles, which are traditionally used in biological experiments, can help to enhance the space charge inhibition ability and increase the electrical breakdown strength of the low-density polyethylene (LDPE) matrix-based nanocomposites [39].

In the present manuscript, polystyrene and polyethylene nanocomposites are prepared by grafting polystyrene (PS) molecules onto the LDPE chains [40]. We call the blend nanocomposite only in a more general sense since PS is chemically bonded to the LDPE backbone. Due to incompatibility, PS and LDPE separate from each other, and PSs exist as nanoscale particles to minimize the interfacial free energy. The distribution of PS particles in LDPE is verified by the scanning electron microscope images.

The PS molecules are grafted onto the LDPE backbones in the low-density polyethylene-g-polystyrene (LDPE-g-PS)/LDPE Composite. Therefore, their homogeneous distribution can be greatly improved. Secondly, the blocking problem can be avoided since the PS molecules go through the finely-meshed filtrating screen together with LDPE at the extrusion temperature, meaning the uninterrupted extrudation of the HVDC cable in the industrial production process. Once more, the graft length and grafting density (the size and the density of the PS particles) can be controlled by the reaction time and monomer-to-initiator ratio. Finally, the β-ray pre-irradiation and suspension grafting methods are suitable for the mass production of nanocomposites. The homogeneous dispersion of the PS nanoparticles can greatly enhance the electrical breakdown strength, and the space charge can be obviously inhibited. The thermally stimulated current (TSC) and conductivity characteristics reveal the increase of deep traps in the nanocomposites and quantum chemical calculation shows PS molecules can play as both electron and hole trapping centers at the same time.

## 2. Material Preparation and Characterization

### 2.1. Preparation of LDPE-g-PS Copolymer

The low-density polyethylene (LD200GH) and styrene monomer (EINECS No. 202-851-5) are manufactured by Sinopec Company Ltd. from Beijing (China) and Fuchen chemical reagent factory from Tianjin (China), respectively. As the first step, LDPE powders were uniformly scattered on the platform trolley and then put into the pre-irradiation apparatus. The *β*-ray pre-irradiation of LDPE powders was carried out in the air at room temperature with a 3 MeV, 120 kW electron beam accelerator and the length of the electron beam is 7.5 cm. The scan width of the electron beam is 1.2 m and the velocity of the platform trolley is 4.8 m/min. The dose of the pre-irradiation was set to be 15 kGy.

The suspension grafting apparatus is composed of a three-necked round bottom flask, polytetrafluoroethylene stirring paddle, water bath kettle, condensation system, thermometer, temperature control system, and mixing control system. The pre-irradiated LDPE (40.3 g), sodium dodecylbenzene sulfonate (SDBS) (3.1 g), water (360 g), benzoyl peroxide (BPO) (0.1%), styrene monomer (40 mL), and hydroxy calcium phosphate (2.5 g) were placed into the three-necked round bottom flask, which is equipped with mechanical stirrer and condenser. The stirring rate was set to be 160 r/min after turning on the blender. The temperature of the temperature control system was set to be 90 °C. In the water bath of 90 °C, the reactants were stirred for 7 h. After the chemical reaction, the product was filtered with a vacuum pump and washed with tetrahydrofuran. Finally, we put the remaining product into the drying oven of 60 °C for 48 h to obtain an LDPE-g-PS copolymer.

The grafting ratio can be introduced to identify the content of the polystyrene:$$GD\left(\mathrm{Grafting}\;\mathrm{ratio}\right)=\frac{W_{g}-W_{\mathrm{LDPE}}}{W_{\mathrm{LDPE}}}\times 100\%$$
where *W*_g_ and *W*_LDPE_ are the mass of the graft product after purification and the mass of the LDPE before grafting, respectively. The grafting ratio measured was 30%.

The corresponding reaction processes are shown in Figure 1. In the macromolecular free radical initiation process, the polyethylene backbone is irradiated in the air to generate active radical sites. Since the irradiation occurs in the air, peroxide radicals are generated due to oxidation of alkyl radicals. Under heating, the hydroperoxides decompose and the macromolecular free radical initiates the graft polymerization in the presence of monomers. For the BPO initiation process, the active sites along the polyethylene backbone are produced by the BPO molecule.

The graft length and grafting density can be technically controlled by the reaction time and monomer-to-initiator ratio. However, it is quite difficult to determine the exact length and density of the graft polystyrene. To roughly estimate the length of the polystyrene chain, we can resort to the electron microscope (SEM) image of the fracture surface of the LDPE-g-PS/LDPE composite, as shown in Section 3.2.

### 2.2. Preparation of LDPE-g-PS/LDPE Blends

All blend samples were prepared by melt blending method. After mixing the LDPE-g-PS copolymer and LDPE with mass ratio 2:8, the 200 mL mixer of the LD-200C torque rheometer, produced by Harbin Hapro Electric Technology Co., Ltd. (Harbin, China) was used to mix the blends for 10 min at 200 °C. After this, the blend was made into sheet samples. In our experiment, the samples were prepared in the press vulcanizer in the following steps: 1. The 80 μm mold with the blend was put into the press vulcanizer and was preheated for 15 min at 150 °C; 2. Then, the pressure was raised three times from 0 to 15 Mp with an increase of 5 Mp each time. For each pressure, the heating plate was kept in place for 5 min; 3. The mold with the sample was taken out and placed in the cooler. The sample was taken out for tests after it cooled down. The pressure is increased from 0 to 15 Mp in three steps in order to avoid any possible air caused defects in the sample. These sheet samples are reserved for further microstructure characterization and electrical tests.

### 2.3. Chemical and Microstructure Characterization

A Jasco FT/IR-6100 Fourier-transform infrared (FTIR) spectrometer (Beijing guojia hengye scientific instrument co. LTD, Beijing, China) was used to obtain the FTIR spectra of the LDPE-g-PS copolymer. Under the transmission mode, the samples were scanned between 400 and 4000 cm^−1^. The background air spectra were subtracted to obtain the final FTIR spectra.

The electrical properties of the nanocomposites depend directly on the dispersion characteristics of the polystyrene nanoparticles in the LDPE matrix. The dispersion condition of the nanoparticles was investigated by SEM Hitachi SU8020 (Beijing science instrument co. LTD, Beijing, China). After being frozen in the liquid nitrogen for 5 min, the sample was broken, and the fracture surface was investigated.

### 2.4. Electrical Properties Characterization


Space charge characterization: The space charge dynamics were characterized using the pulsed electro-acoustic (PEA) method under the applied DC electric field of 40 kV mm^−1^ at room temperature. The sample was 300 μm thick and had aluminum electrodes with a diameter of 25 mm on both sides. Silicone oil was applied between the electrode and the sample to minimize external interference.DC conductance characterization: The E–J curves of the nanocomposites at room temperature were measured by a three-electrode system and samples of 200 μm thick were vapor-deposited with silver to obtain the measuring electrode and guard electrode on one side and the high-voltage electrode on the other side. Then, the three-electrode system was put into the oven (shielding box) and kept for one hour at the chosen temperature (30, 50, and 70 °C) before switching on the circuit. The EST122 picoammeter was used to measure the DC current. The equipment we used to measure the DC conductance is shown in Figure 2. The charging current one hour after switching on the circuit was recorded under the electric field 5–45 kV/mm.DC breakdown characterization: Samples of 100 μm thick were used during the DC breakdown strength test and were vapor-deposited with aluminum to form the aluminum electrodes with a diameter of 25 mm on both sides of the samples. Samples were sandwiched between two columnar electrodes with diameters 25 mm and 75 mm. The DC dielectric breakdown strength was measured at room temperature using a ramping DC voltage with a speed of 1 kV s^−1^. During the measurement process, the whole system was immersed in silicone oil to prevent surface flashover. The Weibull distribution was characterized by the experimental results to obtain the breakdown strength and data dispersiveness.TSC characterization: The samples were set in the vacuum chamber and polarized under a 30 kV/mm DC electric field for 30 min at 60 °C. Then, the samples were quickly cooled down below zero degrees centigrade with liquid nitrogen. Afterwards, the polarization voltage was removed and the samples were short-circuited for 10 min to eliminate the effect of the interface charge. After the short circuit current was less than 1 pA, the TSC was recorded from 0 to 100 °C with a heating rate of 3 °C min^−1^. The TSC measurement system is shown in Figure 3.


## 3. Results and Discussion

### 3.1. FTIR Spectra of LDPE-g-PS

As shown in the reaction processes, the PS molecule is grafted onto the polyethylene backbone by forming C–C or C–O chemical bonds, and it is difficult to identify the chemical bonding between LDPE and PS. After the reaction processes, the product was filtered with a vacuum pump and washed with tetrahydrofuran. PS molecules that are not chemically bonded to LDPE will be dissolved in the tetrahydrofuran solvent. Only PS molecules bonded to LDPE are left in the final product.

In order to identify the PS grafted to the LDPE chains, Fourier-transform infrared spectra (FTIR) was used to check the absorption spectrum of the benzene ring structure on PS. Figure 4 shows the FTIR spectra of PS, LDPE, and LDPE-g-PS. It can be found that new absorption bands appear at 551, 714, 740, 1593, 3010, and 3038 cm^−1^ in LDPE-g-PS compared with the spectra of LDPE. The stretching vibration of =C–H should account for the absorption bands 3038 and 3010 cm^−1^. The absorption peak around 1593 cm^−1^ should be a result of the conjugated C=C stretching vibrations, and =C–H out-of-plane bending and/or ring torsion should contribute to the absorption peaks around 740 and 714 cm^−1^.

### 3.2. Microtopography of LDPE-g-PS/LDPE

LDPE and PS show obvious incompatibility in the cooling process of the LDPE-g-PS/LDPE blends. While LDPE is still in the melting state, PS macromolecules grafted onto LDPE begin to shrink into particles because the crystallization temperature of LDPE is much lower. Figure 5 shows the scanning electron microscope (SEM) image [41,42] of the fracture surface of the LDPE-g-PS/LDPE composite. For pure LDPE, we searched the entire surface of the sample and found no evident particle-shaped structure. Figure 5a shows the surface topography of the pure LDPE sample. For the PS-g-LDPE/LDPE composite, however, we can find an obvious granular structure embedded in the matrix from Figure 5b,c. The diameter of the particle structure is around 50 nm.

Since PS macromolecules are grafted onto LDPE, the obtained PS particles are bonded to LDPE molecular chains. In addition, LDPE-g-PS is very compatible with LDPE in the blending process, leading to the uniform dispersion of PS particles in the composite. The approach of melt-blending LDPE and LDPE-g-PS is an effective way to obtain nanocomposites with nano PS particles, whose size and distribution can be controlled.

### 3.3. DC Electrical Breakdown Strength

The DC electrical breakdown strength was investigated to evaluate the insulation properties of LDPE-g-PS/LDPE nanocomposites. The experimental data were analyzed by Weibull statistics, which is extensively used in determining the breakdown strength of the insulators. The following formula is used:(1)$$P=1-\exp{\left(-E/E_{0}\right)}^{\beta },$$
where *P* is the breakdown probability, *E* is the measured breakdown strength, *E*_0_ is the characteristic breakdown strength at 63.2% cumulative breakdown probability, and *β* is the shape parameter related to the data scatter property. According to the IEC/TC 56, the cumulative breakdown probability is obtained through formula (2) when less than 25 samples are used:(2)$$P_{i}=\left(i-0.5\right)/\left(n+0.25\right)\times 100\%,$$
where *i* is the ordinal number of the samples, which are arranged in increasing sequence according to the breakdown strength and *n* is the total number of the samples, which is 12 in the present manuscript. The results of the DC breakdown tests of both LDPE and LDPE-g-PS/LDPE are shown in Figure 6. The characteristic breakdown strength of LDPE-g-PS/LDPE increases from 358.3 kV/mm to 453.7 kV/mm with an enhancement of 26.6% even though the shape parameter decreases from 11.7 to 9.5.

### 3.4. Space Charge Distribution

The space charge and electric field distribution in LDPE and LDPE-g-PS/LDPE at room temperature were obtained by the PEA method. Figure 7a,b show the space charge in LDPE and LDPE-g-PS/LDPE within 60 min under DC electric field 40 kV/mm. For LDPE, obvious homocharge injection near the cathode occurred, and both the amount and the injection depth of the accumulated homocharge increase with polarization time. The accumulated space charge will lead to the distortion of the electric field. Such serious electric field distortion is very unfavorable for insulation applications and poses serious threats to the operation of HVDC cables. Compared with pure LDPE, LDPE-g-PS/LDPE nanocomposites show much less space charge accumulation in the samples. The homocharge injection is effectively inhibited, and the amount of space charge shows no significant variation with the polarization time. The space charge distribution after the short circuit shows the same characteristics, as shown in Figure 7c,d. On one hand, the space charge amount is obviously reduced in LDPE-g-PS/LDPE nanocomposite. On the other hand, the variation of the space charge with time is relatively small in LDPE-g-PS/LDPE nanocomposite than that in LDPE.

### 3.5. DC Conductance Characterization

The electric field dependent conduction current (E–J) characteristics are measured in order to shed light on the effect of the deeper traps in LDPE-g-PS/LDPE nanocomposites on the charge trapping/detrapping and transport characteristics. In Figure 8, the E–J curves of LDPE and LDPE-g-PS/LDPE are shown, and it is found that both curves can be divided into two stages. For low temperature in Figure 8a, the conduction current density increases linearly with the applied electric field (in the log-log scale) with slopes a bit larger than 1 when the electric field is low. This deviation may be attributed to the increase in the carrier density as the electric field strength increases. In the larger electric field, the slopes for both LDPE and LDPE-g-PS/LDPE get larger. The slopes at lower and higher electric field *k*_1_, *k*_2_, and the critical electric field *E*_c_ are shown in Table 1 for T = 30 °C.

The presence of the critical electric field can be explained by the space-charge-limited current (SCLC) theory [43]. Below the critical electric field, the conduction current follows Ohm’s law:
(3)$$J=e_{0}\mu n_{0}V/d,$$
where *e*_0_, *d*, and *V* are the elementary charge, the sample thickness, and the applied field. *N*_0_ is the thermal equilibrium charge carrier density. The carrier mobility μ is assumed to be independent of the applied field. According to Ohm’s formula (3), the conductivity current density should be proportional to the electric field, i.e., *k*_1_ = 1. As the electric field increases further, carriers in deeper traps take the dominant role and the conduction current obeys the trapped space-charge-limited current relation, given by [44,45]:
(4)$$J=N_{0}\mu e_{0}^{1-l}{\left(\frac{\varepsilon l}{H\left(l+1\right)}\right)}^{l}{\left(\frac{2l+1}{l+1}\right)}^{l+1}\left(\frac{V^{l+1}}{d^{2l+1}}\right),$$
where *N*_0_, *ε*, *H,* and *l* are the effective density of states in the valence band, the permittivity of the sample, the total trap density, and a constant inverse proportional to the measuring temperature.

In the trapped space-charge-limited current formula, *l* is always greater than one, and the exponent of the voltage is always greater than two. It is expected that the Child’s law would be finally satisfied if the electric field or temperature is high enough. As shown in Figure 8b,c, the slopes of the conductivity current curves at higher electrical fields get smaller again and take values around 2. For LDPE-g-PS/LDPE, k_2_ is always larger than that of LDPE, meaning there are more traps in LDPE-g-PS/LDPE composite, which is consistent with the following TSC characteristics and density functional theory calculations.

Space charge distribution in polymer material is closely related to the trapping/detrapping and transport of charge carriers. The thermally stimulated current method is effectively and widely used in characterizing the trap sites of the polymers. The local trapping states in polymers have a dramatic effect on the material’s macro electrical properties. The TSC characteristics of LDPE and LDPE-g-PS/LDPE blend are shown in Figure 9a. Both LDPE and PS are non-polar polymers, and only the trapped carriers contribute to the thermally stimulated current. From the TSC characteristics, it is found that there are more traps in LDPE-g-PS/LDPE composite than in LDPE. The TSC line of LDPE reaches its peak value around 60 °C, showing the release of a large number of trapped carriers. For the LDPE-g-PS/LDPE blend, however, there are two peaks locating at 40 and 77 °C, respectively. As the temperature increases, the trapped charge carriers obtain enough energy to escape the trap centers. The higher thermal stimulated current peak temperature means deeper charge carrier traps in LDPE-g-PS/LDPE. The deeper traps would capture the charge carrier around the electrode–insulator interface and thus increase the charge injection barrier.

To illustrate the trap states in both the LDPE and LDPE-g-PS/LDPE blend, the methodology to depict the trap level based on assumptions of electrons injection and continuous trap level distribution in Ref. [46] is used. The trap level density of pure LDPE and LDPE-g-PS/LDPE blend can be obtained as shown in Figure 9b. The trap level density line of LDPE reaches the peak value of around 0.95 eV, which is consistent with the result of 0.92 eV obtained by Ieda et al. [47]. Comparing with LDPE, both shallower and deeper charge carrier traps have been introduced by the PS in the nanocomposite, which is consistent with the following quantum chemical calculations.

## 4. Quantum Chemical Calculation

The deep trap sites in the nanocomposite can help capture the electrons (or holes) injected from the cathode (or anode). The captured homocharges decrease the efficient electric field at the interface raising the potential barrier for carrier injection. As a result, an effective charge injection blocking is formed. To shed light on the mechanism of the dielectric improvement of the LDPE-g-PS/LDPE composite, the chemical structural energy band (bandgap and trapping level) of LDPE-g-PS is examined by density functional theory (DFT) with SIESTA software [48]. The local density of states (LDOS) integrated over a range of energies around the trap energy level is incorporated to depict the distribution of the trapped charge carriers. The vdW-DF functional of Dion et al. with exchange interaction modified by Berland and Hyldgaard [48] is used to include the van der Waals interaction. The molecular structure is optimized to be the energetically stable state. Figure 10 shows (a) the chemical structure of LDPE-g-PS model, (b) the separate orbital electron energy level, (c) the total orbital electron energy level, (d) and (e) the isosurfaces of local density of states around energy levels, i.e., the lowest unoccupied molecular orbital (LUMO) and the highest occupied molecular orbital (HOMO).

The polystyrene is grafted onto the LDPE molecular. The molecular orbital levels are projected onto the LDPE chain and the grafted polystyrene by recognizing the location of LDOS. For example, the isosurface of LDOS distribution around the highest occupied (HOMO) and the lowest unoccupied (LUMO) molecular orbitals are shown in Figure 10d,e, which characterizes the distribution of charge carrier distribution around the corresponding energy level. It can be found from Figure 10b that the LUMO level of the grafted polystyrene is much lower than that of the LDPE chain, while the HOMO level of the grafted polystyrene is obviously higher than that of the LDPE chain. This means that the grafted polystyrene can supply both electron and hole trapping states. The trapping states lie within 0.03–0.69 eV and 2.06–3.15 eV for the electron and 0.47–2.07 eV for the hole. While the charge carriers (electrons or holes) are traveling in the LDPE matrix, the trapping states on the grafted polystyrene would catch the carriers. The carriers trapped around the grafted polystyrene must overcome the large potential barrier to travel to the next grafted polystyrene, i.e., the next carrier-trapping center. It is interesting to notice that the grafted polystyrene can supply both shallow (<1 eV) and deep (>1 eV) electron traps at the same time, which should be consistent with the TSC experimental results in the previous section. Due to these deep trap sites, the homocharges at the interface are formed for LDPE-g-PS/LDPE composite in contrast to the LDPE case, as shown in the space charge profile after a short circuit in Figure 7. The captured homocharges block the further carrier injection. On the other hand, the addition of the LDPE-g-PS greatly reduces the charge mobility through the deep carrier trapping sites, leading to the inhibition of the ionization and hence the hetero-space charges.

## 5. Conclusions

Polystyrene and polyethylene nanocomposites were prepared by blending LDPE and LDPE-g-PS. The copolymer LDPE-g-PS can be obtained by an electron beam pre-irradiation technique and the suspension grafting method. The nanoscale PS particles enhanced the space charge inhibition ability obviously and the DC electric breakdown strength of the nanocomposite was remarkably improved with an enhancement of 26.6% compared to pure LDPE. Both the DC conductivity and TSC characteristics showed the increase of charge carrier traps. Quantum chemical calculation showed that grafted PS molecules accommodate both shallow and deep trapping states of both electron and hole type, and the trapping states lie within 0.03–0.69 eV and 2.06–3.15 eV for the electron and 0.47–2.07 eV for the hole. Besides the obvious improvement of the dielectric properties, the LDPE-g-PS/LDPE composite circumvents the problem of nano filler’s aggregation, which could lead to the blocking of the finely-meshed filtrating screen in the cable manufacturing process.

## Figures and Tables

**Figure 1 polymers-12-00124-f001:**
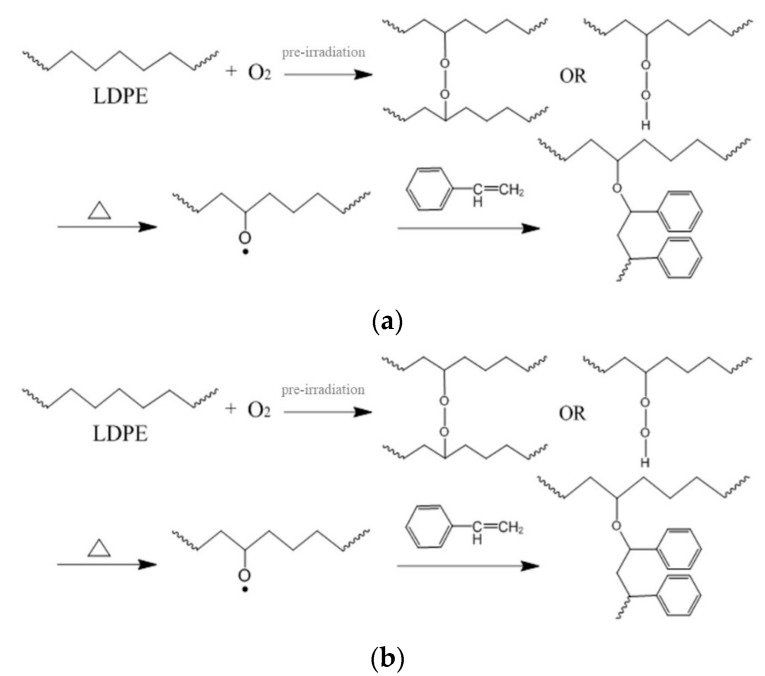
The reaction processes in the production of low-density polyethylene-g-polystyrene (LDPE-g-PS): (**a**) macromolecular free radical initiation and (**b**) benzoyl peroxide (BPO) initiation.

**Figure 2 polymers-12-00124-f002:**
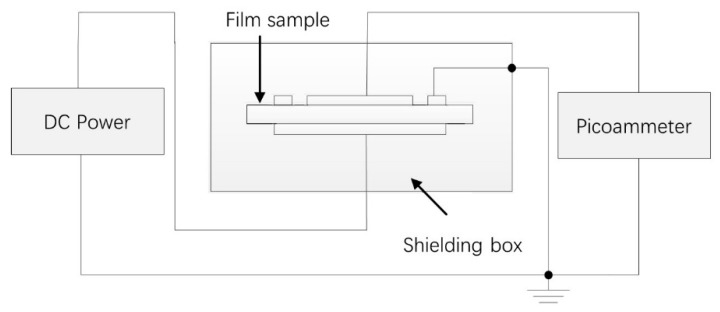
The schematic diagram of the conductivity test system.

**Figure 3 polymers-12-00124-f003:**
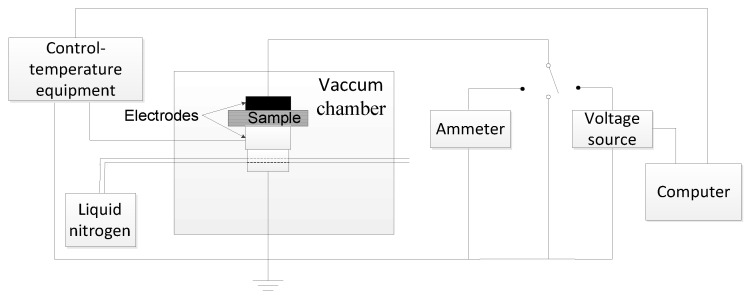
The schematic diagram of the thermally stimulated current measurement system.

**Figure 4 polymers-12-00124-f004:**
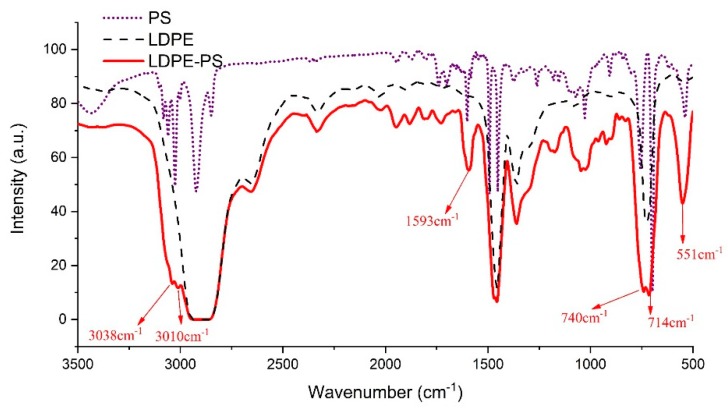
The Fourier-transform infrared spectra of LDPE, PS, and LDPE-g-PS.

**Figure 5 polymers-12-00124-f005:**
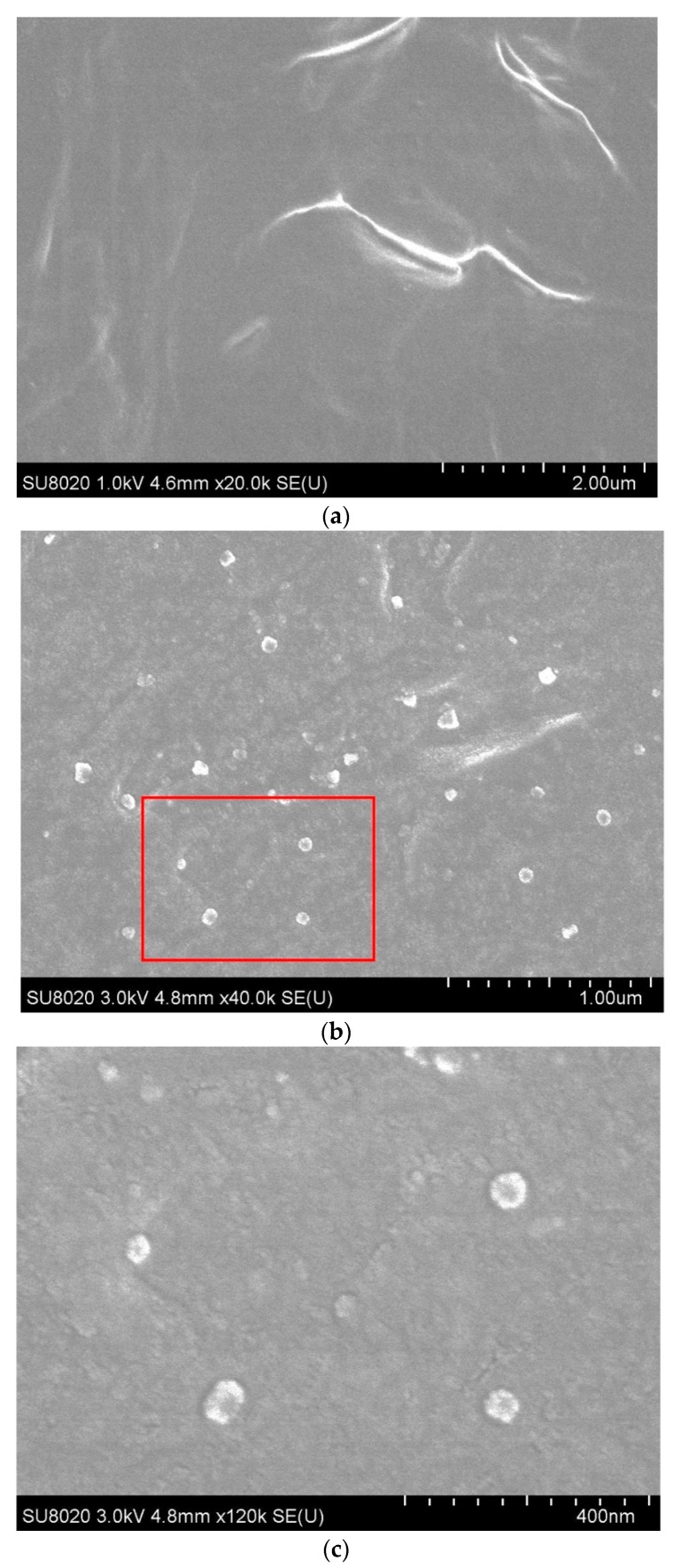
SEM image of the pure LDPE (**a**) and LDPE-g-PS/DPE composite (**b**). The SEM image in (**c**) shows the topography in the red box in (**b**).

**Figure 6 polymers-12-00124-f006:**
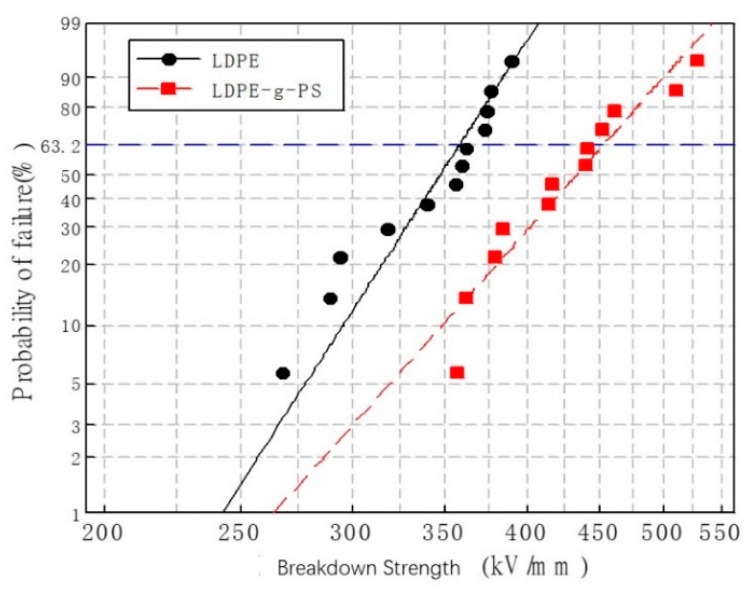
Weibull plots of the direct current (DC) breakdown strength of LDPE and LDPE-g-PS/LDPE blend.

**Figure 7 polymers-12-00124-f007:**
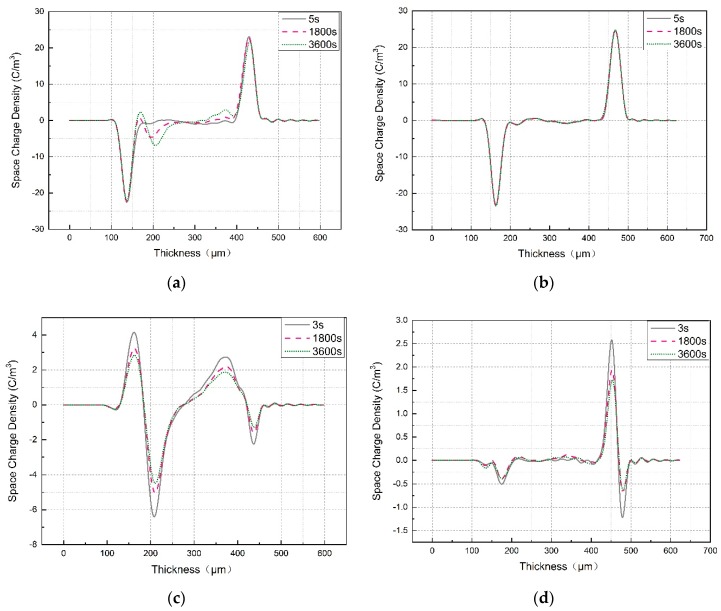
Space charge distribution of LDPE and LDPE-g-PS/LDPE blend (**a**) under applied voltage for LDPE, (**b**) under applied voltage for LDPE/LDPE-g-PS, (**c**) short circuit of LDPE, and (**d**) short circuit of LDPE/LDPE-g-PS.

**Figure 8 polymers-12-00124-f008:**
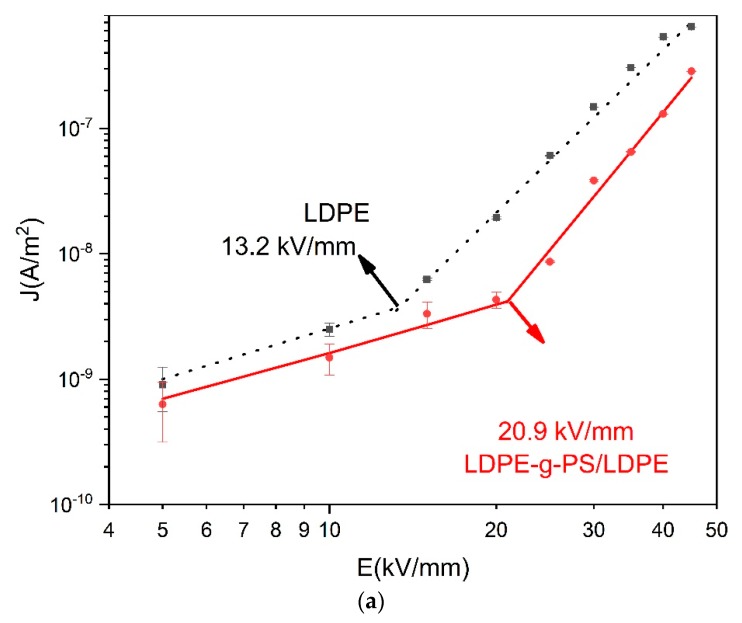
*E–J* plots of LDPE and LDPE-g-PS/LDPE at 30 °C (**a**), 50 °C (**b**), and 70 °C (**c**).

**Figure 9 polymers-12-00124-f009:**
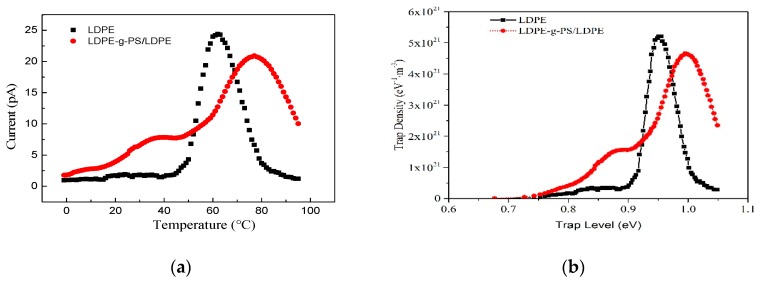
(**a**) TSC lines and (**b**) corresponding trap level distribution of LDPE and LDPE-g-PS/LDPE blend.

**Figure 10 polymers-12-00124-f010:**
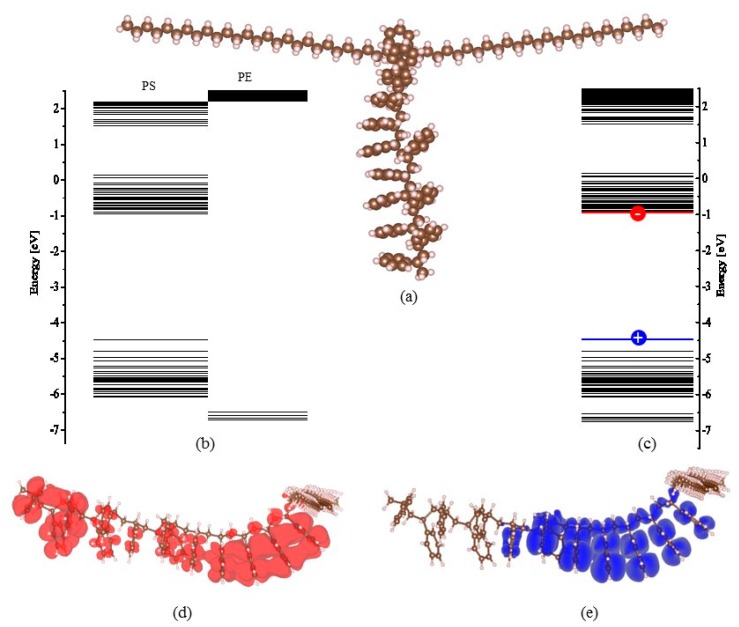
The chemical structure of LDPE-g-PS model (**a**) and the molecular energy levels of PS-g-LDPE molecule (**b**,**c**) and the highest occupied molecular and lowest unoccupied molecular orbitals (**d**,**e**).

**Table 1 polymers-12-00124-t001:** The slopes at lower and higher electric field *k*_1_, *k*_2_ and the critical electric field *E_c_*.

Sample	*k* _1_	*k* _2_	*E_c_*(kV/mm)
low density polyethylene	1.34	4.31	13.2
low-density polyethylene-g-polystyrene/low density polyethylene	1.39	5.38	20.9

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
