# Peer review of "Enhanced Electrical Properties of Polyethylene-Graft-Polystyrene/LDPE Composites"

_polymers, 2020, doi:10.3390/polym12010124_

Round 1
Reviewer 1 Report
The manuscript “enhanced electrical properties of polymer-graft-polystyrene/LDPE composites” is from one point of view a very ambitious work where novel and promising LDPE-graft-PS materials are examined with a multitude of characterization techniques, including among others electrical break down tests, space charge measurements, electrical conductivity measurements and DFT simulations. From another point of view it feels like if the authors were in a hurry when they wrote the paper. It contains a large amount of typos, bad grammar and insufficient explanations. A few examples:
Pg 3: the LDPE and the PS are not examined and described in detail, so it is for instance difficult to know the length of the PS chains and thus indirectly the diameter of the PS “particles”.
Pg 3: You don’t describe how you press the samples. The way of pressing can strongly influence the electrical properties of the sample.
Pg 4: How do you evaporate a sample with an electrode? Check the meaning of the word “evaporate”! What kind of sensor and equipment did you use for the measurements? Why did you measure at room temperature when the typical temperature in a HVDC cable is around 60C?
Pg 4: Did you measure in the oven at different temperatures or did you remove the sample from the oven and measured in room temperature afterwards? Unclear! Write the minimum and maximum field and temperature you used. Write which kind of equipment you used. After how long time did you reach a “steady charging current”?
Pg 5: Can a blend of LDPE and LDPE-g-PS really be called a nanocomposite when the PS “particles” are bonded to the LDPE chains?
The poor standard of the language makes the paper difficult to read and evaluate, but I predict that it will become worth publishing after a major revision. The novelty and the scientific level of the paper is otherwise acceptable. The idea of grafting molecules onto the LDPE chains so that they form spherical “particles” attached to the chains is interesting and useful from an industrial perspective. The improvements, as compared to pure LDPE, are significant, although not extraordinary. However, since the technique is probably easy to up-scale at a low cost, it is clearly promising for HVDC applications.
Author Response
Dear referee,
Please see the attachment.
Kind regards,
Shuwei Song

Reviewer 2 Report
The synthesis of the graft copolymers is not sufficiently described. How does the grafting start? Are there macroradicals or hydroperoxide groups?
There is not a single reference to the grafting reaction and the procedure used. Check for example "Post-irradiation grafting of styrene onto polyethylene" in Radiation Physics and Chemistry 78(2009)521–524.
Was the obtained graft copolymer a thermoplastic or the irradiation caused crosslinking within the PE particles?
Author Response

(The authors gave the same response as above.)

Reviewer 3 Report
The paper under consideration reports electrical properties of polymer composites made of low-density polyethylene (LPDE) grafted with polysterene molecules. The studied polymer blends exhibit improved electrical properties including higher direct current electrical breakdown strength (more than 25% enhancement), and noticeable increase in the charge carrier traps. The obtained results have important practical implications. At the same time, the paper should be improved before recommending it for publications. Some comments are listed below:
Major comments.
Additional evidence is required to prove the existence of LPDE grafted with nanoscale polystyrene particles. Figure 2 should also show an FTIR spectrum of pure polystyrene nanoparticles. The authors should demonstrate the existence of chemical bonds between polystyrene nanoparticles and LPDE (host polymer). The discussion of the morphology of the fabricated polymer composites is not convincing. In fact, the SEM image (Figure 3) can not reveal any nanostructure because of its insufficient magnification. I would recommend using more appropriate SEM images along with pseudo-colors to improve the discussion. The discussion of electrical properties should be improved. Error bars should be shown in Figure 6 (“electric field – current density” plot). Equation (3) accounts only for a single type of charge carriers. In fact, thee are both electrons and holes with different charge mobilities. In addition, the concentration of charge carrier can also be electric field dependent. What is the role of the composition of the blend? What is the reason behind the chosen concentration of components making up the composite?
Minor comments:
Correct numerous typos (Figure 4 – “…stenth”, HOMO and LUMO (page 10) should be swapped).
Figure 8: add (a), (b), (c), (d) to the Figure capture.
Author Response

(The authors gave the same response as above.)

Reviewer 4 Report
Figure 3: SEM image is not clear to me. I think the image does not match the explanation provided. I highly recommend the authors to go for wither AFM or TEM imaging, or perform etching in the SEM. They can refer to the following papers.
Otero-Navas I, Arjmand M, Sundararaj U. Effect of Carbon Nanotube on Morphology Evolution of Polypropylene/Polystyrene Blends: Understanding Molecular Interactions and Carbon Nanotube Migration Mechanisms. RSC Advances, 2017; 7: 54222-54234.
Otero-Navas I, Arjmand M, Sundararaj U. Carbon Nanotube Induced Double Percolation in Polymer Blends: Morphology, Rheology and Broadband Dielectric Properties. Polymer, 2017; 114: 122-134.
Figure 5 should go to section 3.4. The authors should highlight the novelty of the work. Is there an alternative to use a low content of conductivity nanofiller rather than generating a polymer blend system? Why did the authors use 2:8 blend ratio?
Author Response

(The authors gave the same response as above.)

Round 2
Reviewer 3 Report
The authors addressed the received comments reasonably. The paper can be recommended for its publication.
Reviewer 4 Report
The authors have addressed my concerns.
The manuscript merits the publication right now.